# Changes in the Acetylcholinesterase Enzymatic Activity in Tumor Development and Progression

**DOI:** 10.3390/cancers15184629

**Published:** 2023-09-19

**Authors:** Benjamín Pérez-Aguilar, Jens U. Marquardt, Encarnación Muñoz-Delgado, Rosa María López-Durán, María Concepción Gutiérrez-Ruiz, Luis E. Gomez-Quiroz, José Luis Gómez-Olivares

**Affiliations:** 1Area de Medicina Experimental y Traslacional, Departamento de Ciencias de la Salud, Universidad Autónoma Metropolitana, Unidad Iztapalapa, Mexico City 09310, Mexico; benjamin.peag@gmail.com (B.P.-A.); mcgr@xanum.uam.mx (M.C.G.-R.); 2Department of Medicine I, University of Lübeck, 23562 Lübeck, Germany; jens.marquardt@uksh.de; 3Departamento de Bioquímica y Biología Molecular-A, Universidad de Murcia, 30100 Murcia, Spain; encarna@um.es; 4Laboratorio de Biomembranas, Departamento de Ciencias de la Salud, Universidad Autónoma Metropolitana, Unidad Iztapalapa, Mexico City 09310, Mexico; axia@xanum.uam.mx

**Keywords:** acetylcholinesterase, cancer, apoptosis, enzymatic activity

## Abstract

**Simple Summary:**

The cholinergic system’s participation in cancer development has been reviewed, highlighting the involvement of acetylcholine receptors (AChR) and acetylcholine (ACh). It has also been observed that acetylcholinesterase (AChE) plays a relevant role in cancer because AChE is an indirect regulator of AChRs by hydrolyzing ACh; however, controversy has been observed regarding the participation in cancer, since in some tumors the enzymatic activity increases, while in others the activity drops. This review focuses on analyzing the involvement of AChE during cancer progression and proposes AChE as a central regulator in the initiation and progression of cancer via the cholinergic system. Modulating ACh levels with AChE could regulate AChRs differentially, thus driving diverse cancer events.

**Abstract:**

Acetylcholinesterase is a well-known protein because of the relevance of its enzymatic activity in the hydrolysis of acetylcholine in nerve transmission. In addition to the catalytic action, it exerts non-catalytic functions; one is associated with apoptosis, in which acetylcholinesterase could significantly impact the survival and aggressiveness observed in cancer. The participation of AChE as part of the apoptosome could explain the role in tumors, since a lower AChE content would increase cell survival due to poor apoptosome assembly. Likewise, the high Ach content caused by the reduction in enzymatic activity could induce cell survival mediated by the overactivation of acetylcholine receptors (AChR) that activate anti-apoptotic pathways. On the other hand, in tumors in which high enzymatic activity has been observed, AChE could be playing a different role in the aggressiveness of cancer; in this review, we propose that AChE could have a pro-inflammatory role, since the high enzyme content would cause a decrease in ACh, which has also been shown to have anti-inflammatory properties, as discussed in this review. In this review, we analyze the changes that the enzyme could display in different tumors and consider the different levels of regulation that the acetylcholinesterase undergoes in the control of epigenetic changes in the mRNA expression and changes in the enzymatic activity and its molecular forms. We focused on explaining the relationship between acetylcholinesterase expression and its activity in the biology of various tumors. We present up-to-date knowledge regarding this fascinating enzyme that is positioned as a remarkable target for cancer treatment.

## 1. Introduction

### 1.1. Acetylcholine as a Cell Proliferation Factor

Acetylcholine (ACh) is a cholinergic neurotransmitter whose primary function is the chemical synapse [1,2]. Although it has been extensively studied as a neurotransmitter, ACh also has autocrine and paracrine functions related to the promotion of cell proliferation [3,4,5,6,7] and inhibition of apoptosis via acetylcholine receptors (AChRs) [8,9,10,11,12,13,14]. The nicotinic acetylcholine receptor (nAChR) and muscarinic acetylcholine receptor (mAChR) are expressed in tumor cells, and their increment is related to poor prognosis [12,15,16,17]; therefore, strong evidence has suggested that ACh and AChRs are essential regulators in cancer [18].

Activation of nAChRs regulates cell proliferation. It has been found that the scaffolding protein β-arrestin-1 binds and activates Scr in nAChR stimulation via nicotine in non-small-cell lung cancer (NSCLC) and colon cancer cells [19,20]. Furthermore, α7nAChR can activate the MAP kinase pathway [19,21,22], and nicotine treatment of mouse lung epithelial cells shows an upregulation of cyclin-D1 that induces S-phase entry [23]. Similarly, in NSCLC cells, it has been demonstrated that the activations of α7nAChR with nicotine can induce cell proliferation in a manner analogous to the growth factor, causing physical interaction of the retinoblastoma protein (Rb) with Raf-1; this promotes binding of E2F1, E2F2, and E2F3 to the promoter of proliferation-related genes, inducing their transcription and entry into the S-phase of the cell cycle [19]. Moreover, Ca^2+^ influx promoted by α7AChR activates the kinases ERK1/2, MEKK-1, and p90^RSK^; subsequently, p90^RSK^ activates the transcription factor NF-κB, inducing entry into the S-phase in the mesothelial cell line MSTO-211H [21,24,25,26] (Figure 1A).

The apoptotic activity is mediated by α3 and α4nAChRs and involves activation of the Akt pathway [31]; however, other pathways, such as those for PKC, PKA [32], and NF-κB, participate in the apoptotic effects [13]. The activation of the ERK1/2, Akt, and PKA pathways induce phosphorylation of Bad at Ser112, Ser136, and Ser155, respectively, causing its inactivation and preventing cell death [32]. Furthermore, it has been demonstrated that the activation of PKC by nAChRs can induce the phosphorylation of Bax at Ser184, inactivating it while at the same time promoting the phosphorylation of Bcl-2 at Ser70, causing its activation; these events suppress the apoptosis (Figure 1B) [13,14,33,34].

Muscarinic acetylcholine receptors (mAChRs) also activate cell proliferation pathways. Stimulation of the mAChR type 3 (M3) induces the activation of PI3K, adenylate cyclase, phospholipase A_2_ (PLA_2_), and diacylglycerol (DAG) through phospholipase C (PLC) [3,35,36,37]. DAG actives protein kinase C (PKC), regulating MAPk/ERK1/2 through Raf-1 [37,38,39]. This pathway also involves PI3K, Src, and Erk1/2 [39]. Instead, IP_3_ induces the release of Ca^2+^ from intracellular stores, and Ca^2+^ can modulate MAPK/ERK1/2 signaling via Ca^2+^/calmodulin kinase (Ca^2+^/CaM) and Ca^2+^-dependent protein tyrosine kinase (PYK2) [37,40,41,42] (Figure 2A). It is known that these receptors inhibit apoptosis through the activation of PI3K/Akt and MAPK/ERK1/2 in diverse cell types [43,44,45,46,47,48]. The mechanism depends on the transcription of the anti-apoptotic protein Bcl-2 [49]. It was shown that Akt could inhibit both caspase 9 and Bad [50,51,52,53,54,55], and the activation of mAChRs (M1) inhibit caspase 2 and 3 independent of the PI3-K/Akt and MAPK/ERK1/2 pathways [9,10]; therefore, mAChRs have a significant impact on apoptosis (Figure 2B).

These findings suggest a potential role for ACh in cancer and increase the relevance of examining whether ACh might be involved in tumor progression and whether the control of ACh levels is relevant to regulating cancer development. Acetylcholinesterase (AChE) regulates the levels of ACh through the hydrolysis of the neurotransmitter, thereby preventing ACh from reaching the nicotinic acetylcholine receptor (nAChR) [2,56,57] (Figure 3); consequently, AChE could be a direct regulator of these autocrine and paracrine cellular processes by controlling ACh levels.

### 1.2. The Enzyme

In humans, the gene encoding *ACHE* is located on chromosome 7 in the q22 region. In the gene, three exon 1 variants (each with a distinct promoter), five other exons, and a pseudo intron have been identified [58] (Figure 4A). The complex transcription of the gene and post-transcriptional mechanisms can generate up to nine different transcripts [59]. All mRNAs generated sharing exons E2, E3, and E4, which encode for the catalytic domain of AChE [60,61] (Figure 4A) and differ in the 5′ and 3′ exons. Exons E1a and E1c do not have initiation codons, so for the six mRNAs with these codons, translation begins in the E2 exon, and mRNAs differ only in the 3′ end and encode for the three classical isoforms of AChE (T, H, and R) that differ at the C-terminal end [62,63].

The classical AChE-T subunit could generate amphiphilic monomers, dimers, and tetramers (G_1_^A^, G_2_^A,^ and G_4_^A^); hydrophilic tetramers (G_4_^H^); and hetero-oligomeric associations [64]. If the tetramers are bound to the Proline-Rich Membrane Anchor (PRiMA) transmembrane protein, the tetramers are capable of being inserted into the membrane (PRiMA-G_4_^A^) [65]. When one, two, or three tetramers are bound to a collagen-type tail Q (ColQ), the asymmetric forms A_4_, A_8,_ and A_12_ are obtained [66]; these are present in the neuromuscular junctions (Figure 4B)_._

The classical AChE-H subunit produces type I amphiphilic dimers in which each subunit has a covalently linked glycophosphatidylinositol (GPI) moiety [67]. The carboxyl-terminal end of the AChE-R subunit consists of a group of 30 amino acids that lack Cys residues, whereby the subunits remain as monomers [68] (Figure 4B).

However, the E1e exon has an initiation codon, so the three mRNAs generated by alternative splicing at the 3’ end containing the E1e exon produce proteins that have an N-terminal extension; these variants are called N-AChE-(T, H, or R) (Figure 4B). The extension could serve as an anchoring domain of the plasma membrane [69].

## 2. The Implication of Acetylcholinesterase in Tumor Development

Considerable evidence indicates that AChE performs other tasks outside the synapse, including proliferation, cell adhesion, neurogenesis, hematopoiesis, and apoptosis [68,70,71].

In recent years, the role of AChE in the initiation and development of cancer has been extensively studied. Tumors in which mutations in the ACHE gene were most frequently found include ductal, lobular, and tubular breast carcinoma, in which deletions predominate (65.7%) over amplifications (22.9%) [72]. In addition, a relationship between tumor size and amplifications of *ACHE* has been observed [72]. Similarly, Boberg and collaborators found amplification in 62.5% of the sporadic breast cancer samples analyzed, making clear the relationship between the modification in the *ACHE* gene and the development of tumors [73].

In astrocytoma and melanoma, the levels of both mRNA AChE-T and mRNA-R variants increase as the aggressiveness of the tumors rises [74]. On the other hand, in colon cancer, the group led by Dr. Vidal found a decrease in the mRNA levels of AChE-R, AChE-H, and AChE-T [75], whereas in lung cancer, they only found a decrement in the mRNA levels of AChE-T [76]. No changes in expression have been observed in renal carcinoma, skin cancer, or retinoblastoma [77,78]. All this indicates that the expression of AChE in tumors depends on the tumor origin and probably the cancer stage.

The Cancer Genome Atlas (TCGA) contains information on the expression of *ACHE* in diverse tumors that is integrated with clinical characteristics, including patient outcomes. The *ACHE* gene was shown to be downregulated, upregulated, and unchanged in solid tumors (Figure 5); therefore, using UALCAM [79,80] and KMplot [81] web tools, we analyzed the expression profile of the *ACHE* gene at the mRNA level in the five common causes of cancer death worldwide reported by the International Agency for Research on Cancer in 2020 (lung, colorectum, liver, stomach, and breast cancer) [82,83].

In lung adenocarcinoma (LUAD), the expression of *ACHE* is elevated in the primary tumor, and the increase is observed at all stages (Figure 6A). However, there are no significant differences between stages (*p* > 0.05), suggesting that ACHE overexpression is unrelated to tumor progression. Moreover, the survival analysis in the TCGA data set of cancer patients revealed that a low *ACHE* expression correlated with a poor outcome (Figure 7A). In lung squamous cell carcinoma (LUSC), unlike LUAD, the *ACHE* expression did not show changes in the primary tumor at all stages (Figure 6B). However, the survival analysis of the prediction outcome revealed that the high *ACHE* expression was associated with poor prognosis (Figure 7B).

In rectum adenocarcinoma (READ), the expression of *ACHE* decreases in the tumor, and the decrease is independent of the stage in which it is found (Figure 6C). These results suggest that AChE downregulation is unrelated to tumor progression; the survival analysis did not show a relationship between gene expression and prognosis (Figure 7C).

In the case of liver hepatocellular carcinoma (LIHC), the *ACHE* gene was overexpressed in the tumor and significantly increased with the tumor stage (Figure 6D). This trend suggests a relationship between *ACHE* gene overexpression and tumor development. However, the gene expression and prognosis relationship showed no changes between patients with low and high *ACHE* expression (Figure 7D).

For stomach adenocarcinoma (STAD), the results are similar to those of the liver; the *ACHE* gene is overexpressed in the primary tumor, and the increase is significant in stages 2 and 3 (Figure 6E). Furthermore, there are no changes in prognosis between patients with low and high *ACHE* expression (Figure 7E). These data suggest that the upregulation of AChE may be related to progression but not to poor outcomes.

Finally, in breast cancer (BRCA), the *ACHE* gene did not show changes in the primary tumor at any stages (Figure 6F) or in the survival analysis (Figure 7F). These findings indicate that the *ACHE* gene expression in tumors depends on the tumor origin. The gene is overexpressed for tumors such as lung adenocarcinoma, hepatocellular carcinoma, and stomach adenocarcinoma. For the cases of the liver and stomach, the progression of tumors is related to overexpression of the *ACHE* gene, which could suggest that the high AChE content and enzymatic activity have relevant participation in developing these types of tumors. However, in rectal and colon adenocarcinoma, AChE is repressed, and there is no relation between low expression and tumor progression or prognosis.

On the other hand, no changes were found in lung squamous cell carcinoma and breast cancer, indicating that *AChE* expression could not have a relevant role in these tumors. In contrast, the survival analysis showed the opposite results in lung cancer, and other cancers did not show changes between high and low expression of the *ACHE* gene. These data demonstrate that the analysis of tumor progression and patient survival based on *AChE* expression levels does not show convincing results that indicate the relevance of AChE in tumor progression and aggressiveness. However, an analysis of liver hepatocellular carcinoma between the *ACHE* gene expression and poor prognosis showed that a high expression level of AChE was associated with improved overall survival. Patients were divided into three groups: (I) patients with low AChE expression, (II) patients with moderate AChE expression, and (III) patients with high AChE expression. Significant differences were found between groups I and II and between groups I and III, demonstrating that the low expression was associated with a poor outcome [84]. Similar results were found in a cohort of gastric cancer patients [85].

Mutations in the *ACHE* gene and transcriptional regulation must be reflected in the protein synthesis and the catalytic activity. To highlight this point, Castillo-González and coworkers found that tumors with low AChE activity were associated with poorer overall survival, demonstrating that AChE activity could also indicate poor prognosis and have a relevant role during tumor development [86,87]. Therefore, we will focus on analyzing the levels of protein content and changes in the activity in different cancers. The objective is to find a relationship between ACh and AChE and their role in cancer progression.

## 3. Acetylcholinesterase Activity in Tumors

In prostate cancer patients, Battisti and coworkers observed a decrease in AChE activity in the blood [88]; this opened the possibility of proposing AChE as a cancer biomarker. On the other hand, Nieto-Cerón and collaborators reported no significant changes in AChE activity between prostate cancer and benign prostatic hyperplasia [89]; however, they analyzed two types of tumors: metastatic and non-metastatic. These data indicate that there are no enzymatic changes between different stages of cell differentiation; therefore, it would be essential to analyze changes in the molecular form of AChE in different stages to analyze whether changes occur at this level.

A decrement in the AChE activity has been observed in other tumors (Table 1); for example, in lung cancer [76,90] and colorectal carcinoma, where a decrease was found in the expression of AChE-T, AChE-H, and AChE-R [75], and this was correlated with a decrement in AChE activity. Also, a reduction in enzyme activity has been reported in gliomas, and interestingly AChE tetramers contain a monomer of butyrylcholinesterase (BuChE) [91]. Finally, in hepatic tumors [84,92], gastric cancer [85], and head and neck carcinomas [86,87], a significant decrease in AChE activity was found. Interestingly, in the head and neck, squamous cell carcinoma was associated with low AChE activity and poorer overall survival [86,87]. In addition, some works report alterations in the *ACHE* gene, such as in breast cancer; these alterations decreased the enzymatic activity in axillary lymph nodes, which was heightened as the malignancy of the tumors progresses [93].

Regarding this phenomenon in breast cancer, Dr. Garrido’s group demonstrated in the rat mammary gland that inhibition of AChE with physostigmine (eserine) induced carcinogenesis, showing a relationship between the decrease in AChE activity and carcinogenesis [94]. On the other hand, Zhao and coworkers demonstrated in hepatocellular carcinoma (HCC) patients that the expression was related to overall survival. They found that high expression of AChE improved overall survival in patients with HCC and that low AChE expression was an indicator of poor overall survival [84]. Similarly, gastric cancer patients with high AChE levels also demonstrated better overall survival, and the low AChE expression decreased overall survival [85].

There is a clear relationship between low AChE activity and poor prognosis in cancer patients; however, it is relevant to figure out why a reduction in AChE activity and protein content induces tumor progression and aggressiveness.

### 3.1. The Possible Role of Acetylcholinesterase in Tumors with Low Enzymatic Activity

The low AChE activity in tumors of diverse origins and the contribution of this protein in apoptosis could be related to the control of ACh levels and the alternative function [70,71].

Regarding the participation of the AChE in apoptosis, Zhang and coworkers reported that inducing apoptosis via different stimuli increased the expression of AChE-T mRNA [70], suggesting a relationship between the synthesis and the catalytic activity of AChE with this process. In H_2_O_2_-induced apoptosis, *ACHE* overexpression is mediated by JNK signaling and the activation of the transcription factors AP-1 and ATF2 [71].

In PC12 neuroendocrine cells, the induction of the cytoplasmic Ca^2+^-dependent apoptosis promoted the overexpression of AChE-T through glycogen synthase kinase β (GSK3-β) [95]. Transfection of the N-extended variant of AChE-T in embryonic kidney cells (HEK 2963), glioblastoma (U87MG), lung epithelium (T84), hamster ovary (CHO) cells, and primary cortical cells induced apoptosis via a mechanism dependent on GSK3-β [96]. These data demonstrated a relationship between apoptosis and an increment in the synthesis and enzymatic activity of AChE. In cancer, a decrease in AChE activity could be (but is not necessarily) due to a reduction in the synthesis of this protein, thus avoiding apoptosis; however, the question is: what is the role of the AChE in apoptosis?

Park and collaborators proposed that AChE protein participates in apoptosome formation [97,98]; they showed via a co-immunoprecipitation assay that AChE interacts with Apaf-1 and cytochrome c to induce apoptosis. The interaction between AChE with Apaf-1 and cytochrome c disappeared under the influence of a specific siRNA for AChE, and apoptosis was significantly decreased [97]. They also demonstrated that AChE interacts with caveolin-1 before binding to cytochrome c and Apaf-1; if the interactions are prevented, the apoptosome formation is abolished (Figure 8) [98].

These findings place AChE as a central regulator of apoptosis. At the same time, the deregulation and low content could highlight the participation in carcinogenesis and cancer progression. These findings position AChE as a possible tumor-suppressor protein because the apoptosis could be reduced, providing cancer cells with more remarkable survival. It is essential to analyze whether enzymatic activity is also relevant in cancer. In this case, it is crucial to explore the ACh levels and the relationship with the AChE activity.

ACh promotes cell proliferation and inhibits apoptosis through muscarinic and nicotinic receptors and a well-known signaling pathway (Figure 1 and Figure 2). The decrement in AChE activity could increase the local levels of ACh; this could contribute to tumor development (Figure 3) [76,84,99].

In this regard, Professor Roskams’s group demonstrated the activation of progenitor cells through a mechanism dependent on muscarinic type 3 receptor activation in hepatic vagotomy in rats. In partial hepatectomy, the AChE content decreased, allowing ACh to reach to its target in oval cells to promote proliferation and repair liver damage [100]. Based on these data, it could be assumed that the decrease in AChE activity and the increase in the local concentrations of ACh could contribute to tumor aggressivity.

In human lung tumors, high ACh levels are related to low AChE activity [76]; similar findings can be found in human liver tumors and HCC cell lines, indicating that ACh levels are inversely correlated with AChE activity [84,92]. The elevation of ACh concentrations could also be relevant to migration and invasion processes [101]. Our group has demonstrated that the inhibition of AChE increases the cell proliferation rate and the sphere-forming capacity in the HCC cell lines HepG2 and Huh-7 cells; this effect was potentiated by the addition of ACh or an AChE inhibitor; therefore, the enzyme activity, which has been shown to decrease in liver tumors, plays a significant role in the progression of hepatic tumors because it is not enough to deactivate ACh (Figure 9) [92]. These findings demonstrate that the low AChE activity could be strongly related to carcinogenesis and tumor aggressivity.

Although it is known that the AChE protein could be a potential tumor suppressor [97,98], it is imperative to elucidate the mechanism that decreases the synthesis and activity. It has been shown that exposing the pheochromocytoma cell line PC12 to H_2_O_2_ decreased Akt phosphorylation and AChE activity increments; this was associated with cytochrome c cytosolic release and caspase activation [102]. Nevertheless, when the Nerve Growth Factor (NGF) and H_2_O_2_ were added, Akt activation remained constant, showing decreased AChE activity and reversing apoptosis [102]. This result suggests a relationship between the PI3k/Akt signaling pathway and the downregulation of AChE activity. This is relevant because the overactivation of the PI3k/Akt pathway is a frequent event in human cancers [103,104]; therefore, we could hypothesize that the hyperactivation of PI3k/Akt could decrease AChE content; this event affects the apoptosome formation and enzymatic activity, allowing high ACh levels that promote cell proliferation and inhibition of apoptosis through ACh receptors (AChRs) (Figure 10).

One of the primary mechanisms that sustain the activation of the PI3K/Akt pathway is the silencing of the lipid phosphatase PTEN; this enzyme dephosphorylates phosphatidylinositol 3,4,5-trisphosphate (PtdInsP_3_), the primary activator of Akt. A relationship has been observed between the high content of the DNA methyl transferase 1 (DNMT1) and the hypermethylation of the promoter region of the *Pten* gene, causing a low gene expression [105,106]; this promotes a constant activation of the PI3k/Akt pathway and could cause a decrease in AChE content. However, another mechanism that could also participate in the regulation of AChE is the direct epigenetic control; there is evidence that epigenetic mechanisms could regulate AChE. It has been documented that in the regulation of AChE by the DNMT1, the use of 5-Aza-2’-deoxycytidine, a DNMT1 inhibitor, restores the levels of AChE content downregulated in HCC cells [107]. This is relevant because DNMT1 is modulated in other cancers such as pancreatic, gastric, ovarian, brain, lung, and liver [108,109,110,111,112,113,114,115].

The AChE also regulates cell proliferation. In the hepatoblastoma cell line, HepG2 treated with ciglitazone (an agonist of the nuclear peroxisome proliferator-activated receptor-gamma) significantly increased the AChE content, inhibiting the cell cycle [116]; this is supported by other works that provided the same evidence of arresting the cell cycle in the G2/M-phase [84,117,118]. The possible mechanism of cell cycle arrest proposed for HCC has been related to a downregulation of the MAPK and Akt pathways, the activation of GSK3β, the degradation of β-catenin, and the suppression of cyclin D1 [84] (Figure 11). These events allow us to hypothesize that the blockade of the PI3K/Akt pathway causes an increase in AChE content and enzymatic activity. Then, a reduction in ACh levels affects cell proliferation.

AChE has different molecular forms; therefore, analyzing which specific molecular forms mediate in these processes is relevant. Although the evidence indicates that AChE-T is the one that participates in the processes of apoptosis, the mRNA of this isoform encodes a wide variety of molecular forms.

#### Patterns of AChE Molecular Forms Expressed in Tumors with Low Enzymatic Activity

Analyses of molecular forms of AChE have been performed in some neoplasms and adjacent healthy tissues to determine whether there were differences in the profile of the molecular forms (Table 1). Could there be a relationship between the molecular forms of AChE and the regulation of proliferation and apoptosis in cancer?

**Table 1 cancers-15-04629-t001:** Status of the acetylcholinesterase (AChE) forms in different tumors.

Tumor Type	AChE Activity	Molecular Form	Reference(s)
Lung cancer	Low	G_1_^A^ and G_2_^A^	[76,90]
	High	-	-
Colon cancer	Low	G_1_^A^ and G_2_^A^ G_2_^H^, G_4_^A^, G_4_^H^, and A_4_ (disappeared)	[75]
	High	-	[119]
Brain tumor	Low	G_4_ contains butyrylcholinesterase monomers	[91]
	High	NA	[120]
Liver cancer	Low	NA	[84,92]
	High	-	-
Gastric cancer	Low	NA	[85]
	High	-	-
Breast cancer	Low	G_1_^A^ and G_2_^A &^ G_4_^H^, A_4_ y A_8_ (disappeared) ^&^	[93] ^&^, [94]
	High	G_1_^A^ (low content) and G_2_^A^	[121]
Head and neck carcinoma	Low	G_1_^A^ and G_2_^A^ (no enzymatic change) G_4_^A^ (disappeared)	[86,87]
	High	-	-
Renal cancer	Low	-	-
	High	G_1_^A^ (no change) and G_2_^A^ (high activity)	[77,122]
Pancreatic cancer	Low	-	-
	High	NA	[123]

Abbreviations: G_1_^A^: amphiphilic monomers; G_2_^H^: hydrophilic monomers; G_2_^A^: amphiphilic dimers; G_4_^A^: amphiphilic tetramers; G_4_^H^: hydrophilic tetramers; A_4_: asymmetric form with one tetramer; A_8_: asymmetric form with two tetramers. ^&^: Axillary lymph nodes; metastasis from breast cancer.

Martínez-Moreno and coworkers showed a decrease in mRNA-T expression in human lung cancer compared with the other two mRNA types of AChE (H and R) [76]. This effect should be reflected in the expression of the molecular forms. The molecular components found in non-cancerous systems were mainly the dimers (G_2_^A^) and, to a lesser extent, the amphiphilic monomers (G_1_^A^) [76]; the same results were found in lung carcinoma [76]. Although patterns of AChE forms were not affected by cancer, a decrease in the total activity of AChE in both molecular forms was observed [76]. Changes in the monomers’ expression may favor these lung tumors’ survival. Monomers could be responsible for the interaction with caveolin-1, which later participates in apoptosome formation. The AChE must enter the nucleus and leave this compartment to join the apoptosome [98]. Therefore, the monomers could be the leading candidates to participate in this phenomenon. Subsequent studies have shown that lung cancer cell lines (as well as lung tissue) presented patterns of AChE dimers and monomers [90], pointing out that the variations in these two molecular forms could contribute to the survival of the tumors.

In colon cancer patients, Dr. Vidal’s group demonstrated that several molecular forms of AChE were present in healthy colon tissues: G_1_^A^, G_2_^A^, G_2_^H^, G_4_^A^, G_4_^H^, and A_4_. When they analyzed cancerous tissues, they only found G_1_^A^ and G_2_^A^, observing a change and noticing a decrease in the monomers [75]. Therefore, as in cancerous lung tissues, the monomers change, which could favor the survival of the tumors.

A similar effect was observed in breast cancer metastasis. Dr. Vidal’s group analyzed normal axillary lymph nodes and axillary lymph nodes with breast metastasis. In normal tissues, the molecular forms determined were G_1_^A^, G_2_^A^, G_4_^H^, A_4_ and A_8_, which showed an essential change in the cancerous tissues, since only the presence of G_1_^A^ and G_2_^A^ forms was observed along with a decrease in the monomers [93].

In head and neck squamous cell carcinoma patients, G_1_^A^ and G_2_^A^ molecular forms were found with enzymatic activity comparable to normal tissues (except for G_4_^A^, the absence of which was noted in cancerous tissues) [86,87].

In the cases of colon cancer, metastasis from breast cancer to axillary lymph nodes, and head and neck squamous cell carcinoma, there are changes in the patterns of molecular forms that affect all enzymatic forms. However, it affects the molecular forms with a greater hydrolytic capacity, such as tetramers and asymmetric forms. Therefore, this decrease could not only affect the processes of apoptosis mediated by AChE, but these substantial changes in the molecular forms could allow a local increase in the amount of ACh, favoring the cell proliferation and inhibition of apoptosis processes. It can be assumed that the changes in the patterns of the molecular forms—either by decreasing their activity or changing their proportions—would promote cell proliferation and cancer cells’ survival.

### 3.2. Enhanced Enzymatic Acetylcholinesterase Activity in Tumors

An increase in the enzyme has also been found in other types of tumors (Table 1). In brain tumors, it has been found that AChE activity increases considerably as tumor malignancy progresses [120]. Similar findings have been observed in renal [122], breast [121], and pancreatic tumors [123]. These observations oppose the proposed function for the AChE as a potential tumor suppressor. However, AChE also participates in cell adhesion [124], a key event in metastasis.

#### 3.2.1. The Possible Function of Acetylcholinesterase in Tumors with High Enzymatic Activity

If AChE participates in cellular adhesion in tumors with more enzymatic activity, AChE could favor metastasis. It is important to emphasize that based on neuroblastoma observations, Johnson and collaborators proposed that the molecular forms of AChE anchored to the plasma membrane G_4_^A^ are involved in the cell adhesion process. The peripheral anionic site of the AChE is the zone involved in the mechanism. Therefore, high AChE content could affect neuroblastoma development [125,126]. On the other hand, Dr. Weber’s group has shown that in colon cancer, there is an increase in the AChE activity and the binding of c-Myb to DNA accompanied by the cell adhesion; they showed that AChE influences intracellular signal transduction [119]. This is important because the family of the Myb transcription factor is involved in cell adhesion to fibronectin [127]. These data suggest that the increase in AChE content could be related to metastasis.

Previously, we analyzed a relationship between high levels of ACh and low levels of AChE activity and how this balance participates in cancer progression and aggressiveness because the ACh induces cell proliferation, inhibition of apoptosis, invasion, and migration. However, we can hypothesize an opposite balance in which there is high AChE activity and low levels of ACh; therefore, the ACh effects could not take place, and the cancer progression and aggressiveness would have to slow down. However, in these tumors, high AChE activity is related to cancer progression and aggressiveness.

Through the receptor α7nAChR, ACh displays anti-inflammatory properties in immune cells and regulates immune responses [128,129,130,131]. It is known that inflammation has a role during tumorigenesis (from tumor initiation to progression to metastases); therefore, an inflammatory microenvironment is a critical component in all tumors [132,133], and ACh could be a relevant participant during tumorigenesis. It has been observed that the autonomous nervous system impacts cancer. The high vagal activity predicted a better prognosis in colon, non-small lung, prostate, and breast cancer patients [134,135,136]. Vagotomy has been related to an increased risk of gastric, colorectal, prostate, and lung cancers [137,138,139,140]; therefore, high levels of ACh could be participating in generating an improved prognosis in cancer by decreasing the inflammation. However, in a tumor with high AChE activity, these effects could decline due to the high amount of hydrolysis of ACh. Based on this, the high levels of AChE activity in the tumors could disrupt the anti-inflammatory properties of ACh; hence, the ACh does not reach immune cells. This phenomenon could allow inflammation increases. In this context, in a developing tumor, macrophages are the most abundant immune cells (tumor-associated macrophages (TAMs)) at all tumor stages. We can find classically activated (M1) and alternatively activated (M2) macrophages. M1 macrophages display anti-tumoral properties, while M2 macrophages are associated with a pro-tumoral role because they fail to eliminate tumoral cells and contribute to cancer progression; these latter macrophages resemble TAMs [141,142,143], and a high TAM content correlates with poor prognosis [144]. Dr. Demir’s group demonstrated that in pancreatic cancer with high AChE expression, the inhibition of AChE or the administration of ACh reduced both TAM infiltration and serum pro-inflammatory cytokine levels [123]. Therefore, high levels of ACh could reduce TAM and hence decrease cancer initiation, progression, and metastasis. Clinical results have demonstrated that the expression of α7nAChR in TAMs was associated with a low incidence of liver metastasis in colorectal cancer patients and that α7nAChR knockdown in THP-derived macrophages increases the migration and invasion of colorectal cancer cells [145]. These findings demonstrated an indirect role of AChE activity during cancer development, indicating that high enzymatic activity is relevant to induce inflammation and an increment in TAMS through a reduction in ACh levels. These events are related to poor prognosis in cancer; therefore, the high AChE activity could favor an environment that is positive for cancer development and promotes metastasis.

#### 3.2.2. Patterns of Molecular Forms Expressed in Tumors with High Enzymatic Activity

It has been proposed that the molecular form G_4_ participates in cell adhesion processes [125,126]. AChE activity in breast tumors has been detected compared with the control. The molecular forms found were G_1_^A^ and G_2_^A^, which had a lower content of monomers [121]. Similarly, G_1_^A^ and G_2_^A^ molecular forms have been described in renal tumors. Although no changes were found in molecular form patterns, the increase in the enzymatic activity in these tumors was due to the dimers anchored to the cellular membrane by a glycosylphosphatidylinositol (GPI) and not due to the monomers [77,122]. In this regard, it can be assumed that in tumors with high AChE activity and tumors with low enzymatic activity, the monomers decrease to evade programmed cell death. However, these tumors showed a high enzymatic activity governed by a high content of dimers anchored to the membrane by the GPI anchor. It could be assumed that a high dimer content regulates the local ACh concentration, allowing the inflammation to increase and simultaneously realizing the cell adhesion function, since this process may not be unique to tetramers; therefore, the membrane-anchored dimers by GPI could perfectly perform this function. On the other hand, the low content of the monomers could favor tumoral cells’ survival because it is not sufficient for the formation of the apoptosome. Therefore, the high dimer content decreases the local ACh concentration, hence the inflammation increase, while the low monomer content increases the tumors’ survival because the apoptosome cannot form.

## 4. Conclusions

We reviewed the current evidence supporting the participation of AChE in developing different types of tumors. Changes in the activity of AChE favor cancer progression. In this regard, tumors with differential AChE activities coincide at a critical point to decrease the content of the amphiphilic monomer. This implies that changes in the amphiphilic monomer content could favor or impair apoptosome formation; reducing monomer content could cause the tumors’ high survival. That is why it would be relevant to prove the participation of monomers in the apoptosome formation, since the monomers could (via intracellular signals for programmed cell death) avoid the exocytosis pathway and pass from the endoplasmic reticulum to the cytosol through the ERAD pathway (the endoplasmic reticulum-associated degradation system) [146,147].

It was demonstrated that AChE activity has essential participation in cancer. High AChE activity in tumors affects the tumoral microenvironment because of ACh hydrolysis, and inflammation increases, favoring tumor development. Low AChE activity in tumors directly affects the tumor rather than the microenvironment. Low AChE activity increases the local ACh concentrations, causing tumoral growth, aggressiveness, and metastasis.

Although these findings were found in different cancer types and tumoral stages, it led us to formulate a hypothesis involving AChE from tumor initiation to progression to metastasis. We could hypothesize that in the initial cancer stages, initiated cells could have high AChE activity, diminishing the local ACh concentrations, causing both the inflammation and TAMs to increase for a long time and resulting in the appearance of tumors. As tumors develop, enzyme activity levels and AChE content could decrease due to the overactivation of the PI3k/Akt survival pathway and overexpression of DNMT1. This phenomenon could cause an increment in the local ACh concentrations, which could promote proliferation (tumoral growth), inhibition of apoptosis (survival), migration, and invasion (metastasis). This could leave clear the relevant participation of the AChE in cancer (Figure 12).

In addition to the classic function of AChE in nerve and neuromuscular junctions, the alternative roles that AChE develops are highly transcendental. We were able to demonstrate this in different types of tumors. The participation of this protein in the molecular processes involved in the development and maintenance of a cell-proliferation altered state remains to be established and explained in this field through arduous efforts by different research groups worldwide. Although the general idea is that AChE is a well-known and studied protein, the molecular bases of these new functions are still unknown, making AChE a multifunctional protein in living systems.

## Figures and Tables

**Figure 1 cancers-15-04629-f001:**
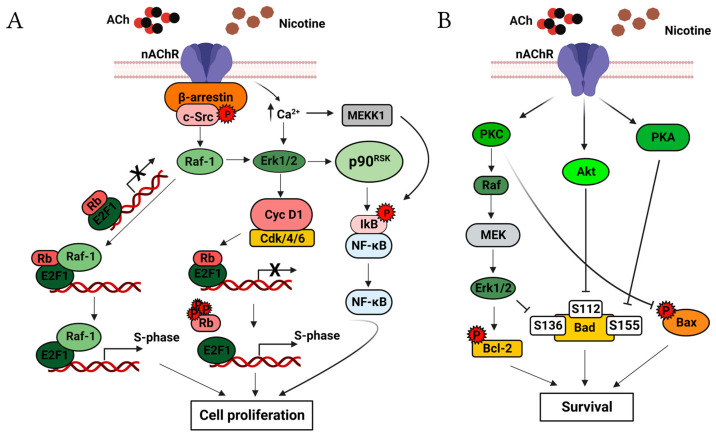
Selected nAChR-mediated proliferative and apoptosis signaling pathways. (**A**) Binding of ACh or nicotine to nAChRs induces the formation of the oligomeric complex, which consists of nAChR, β-arrestin, and Src; this complex activates Src. Activated Src triggers the MAP kinase pathway and induces the formation of the cyclin D1-Cdk4/6 complex, leading to the phosphorylation of Rb [23,27,28]. The hyperphosphorylation of Rb releases the transcription factor E2F1 on proliferation-related gene promotors, leading to S-phase entry. The MAP kinase pathway also induces binding between the Rb-E2F1 dimer and Raf-1. The sustained mitogenic signaling leads to the dissociation of Raf-1 and Rb, leaving free E2F1. The increased influx of Ca^2+^ by nAChRs causes ERK1/2 and MEKK1 activation, MEKK1 activates NF-κB, and cell proliferation is induced [21,24,25,26,29,30]. (**B**) nAChR causes phosphorylation of Bad and Bax, thereby inactivating them and preventing apoptosis. Overexpression of Bcl-2 and its activation by nAChRs induces cell survival. The PKC, Akt, PKA, and MAP kinase pathways mediate this signaling. Created with Biorender.com (accessed on 12 June 2023).

**Figure 2 cancers-15-04629-f002:**
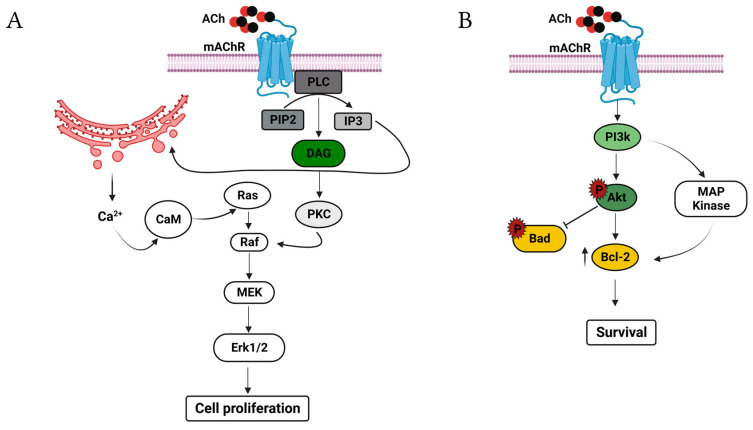
Selected mAChR-mediated proliferative and apoptosis signaling pathways. (**A**) Binding of ACh actives PLC, which hydrolyzes PIP2 to IP3 and DAG. Subsequently, DAG activates PKC, and PKC induces the activation of MAPK to increase DNA synthesis. Calcium mobilizing from organelle stores also activates MAP kinases via CaM, inducing cell proliferation. (**B**) Activation of the mAChRs induces the phosphorylation of Akt by PI3K; Akt promotes survival through the increment in Bcl-2 and inactivation of Bad. PI3K also increases the levels of Bcl-2 through the activation of MAP kinase, inducing cell survival. Created with Biorender.com (accessed on 12 June 2023).

**Figure 3 cancers-15-04629-f003:**
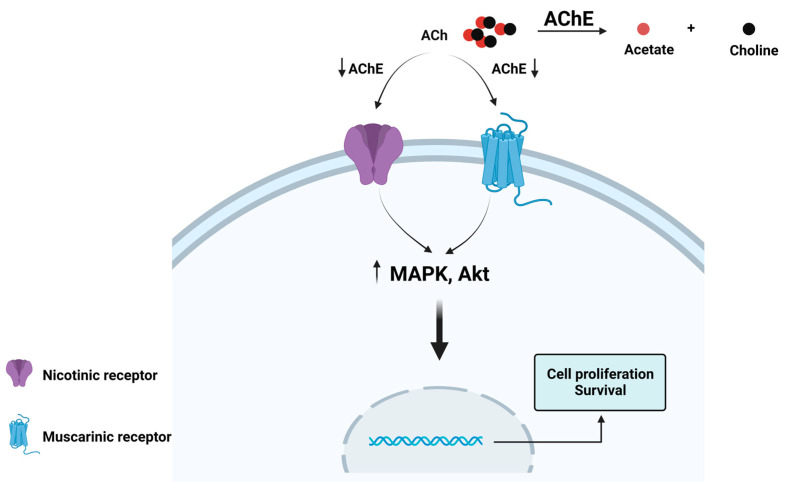
Hydrolysis of acetylcholine (ACh) by acetylcholinesterase (AChE). AChE is a serine hydrolase that rapidly cleaves ACh in acetate and choline. It is on the list of enzymes classified according to their catalytic efficiency, occupying the second position. AChE functions at the limit of substrate diffusion, and it can hydrolyze 25,000 molecules of ACh per second [2]. A decrement in the AChE activity could allow ACh increases and reach nAChRs and mAChRs, promoting proliferation and survival through the MAPK and Akt pathways. Created with Biorender.com (accessed on 12 June 2023).

**Figure 4 cancers-15-04629-f004:**
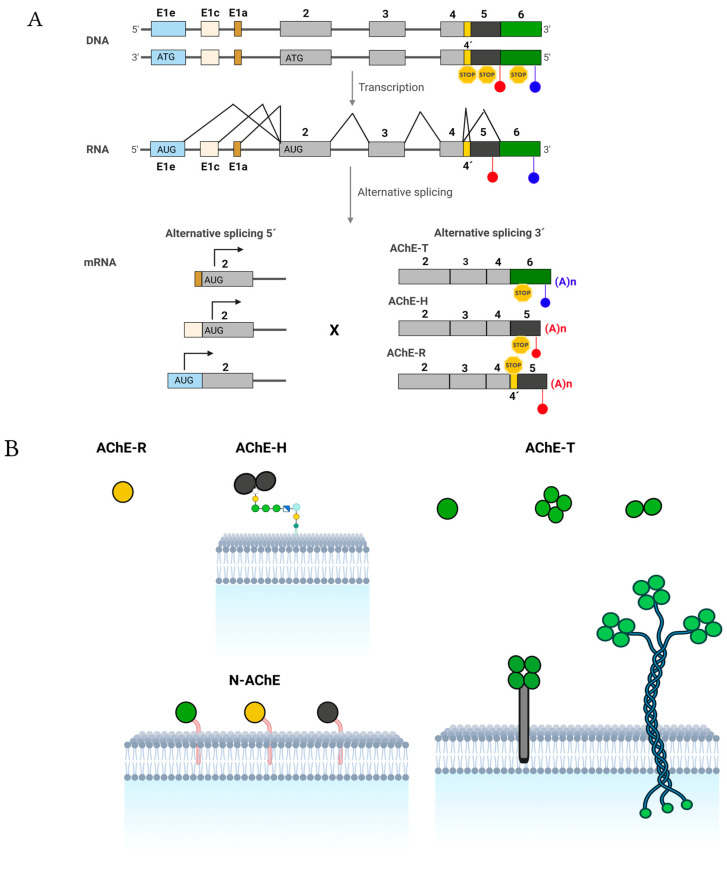
Structure and expression of the human acetylcholinesterase gene, molecular forms of the AChE protein originated by oligomerizing the various subunits, and quaternary associations of other proteins. (**A**) From 5′ to 3′, the ACHE gene has three exon 1 variants (E1e, E1c, and E1a, each with a distinct promoter), three common exons (E2, E3, and E4, shown in gray) with all the information necessary to produce catalytically active proteins, a pseudo intron (I4′, shown in yellow), and two exons (5 and 6, shown in black and green, respectively). Introns are shown as bold solid lines, and light lines connecting exons mean constitutive and alternative splicing. PolyA sites (blue and red circles), initiation (ATG), and STOP codons (yellow hexagons) are also highlighted. Via transcription, three different 5′ ends can be produced by activating different E1 promoters (alternative 5′ splicing), and three other 3′ ends can be generated by alternative splicing of I4′, E5, and E6. Therefore, nine types of mature mRNAs can be produced from AChE. For the six AChE mRNAs with exons E1c or E1a that lack an initiation codon, translation begins at E2 and generates the identical isoforms of AChE regardless of the E1 used. Therefore, three isoforms of AChE may differ at the C-terminal end (the classical isoforms AChE-T, AChE-H, and AChE-R). The translation begins in this exon for the three AChE mRNAs with exon E1e, which has an initiation codon. The three isoforms generated will contain an extension N-terminal (the N-extended isoforms N-AChE-T, N-AChE-H, and N-AChE-R). (**B**) AChE-R consists of monomeric variants. AChE-H or AChE-E (hydrophobic or erythrocytic AChE) produces amphiphilic membrane-anchored monomers and dimers via GPI. AChE-T or AChE-S (synaptic AChE) generates globular components that may or may not have the PRiMA membrane-binding protein or the collagen-type Q tail (ColQ) (asymmetric form). The N-AChE variants are anchored to the membrane through the N-terminal generated by exon 1Ee, N-AChE-T (green), N-AChE-R (yellow), and N-AChE-H (gray). Adapted from [62,63]. Created with Biorender.com (accessed on 12 June 2023).

**Figure 5 cancers-15-04629-f005:**
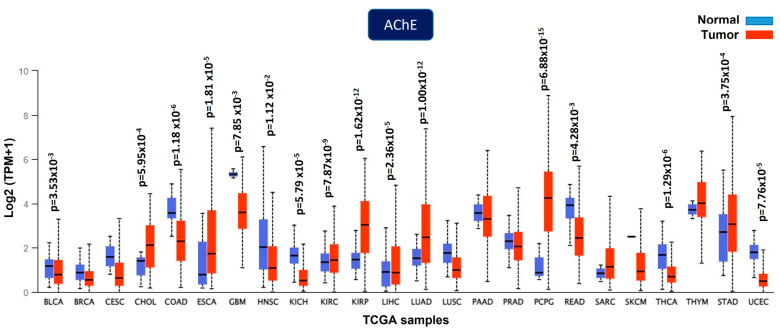
AChE is dysregulated in several solid tumors. Downregulation was found in bladder urothelial carcinoma (BLCA), colon adenocarcinoma (COAD), glioblastoma multiforme (GBM), head and neck squamous cell carcinoma (HNSC), kidney chromophobe (KICH), rectum adenocarcinoma (READ), thyroid carcinoma (THCA), and uterine corpus endometrial carcinoma (UCEC). Upregulation was found in cholangiocarcinoma (CHOL), esophageal carcinoma (ESCA), kidney renal clear cell carcinoma (KIRC), kidney renal papillary cell carcinoma (KIRP), liver hepatocellular carcinoma (LIHC), lung adenocarcinoma (LUAD), pheochromocytoma and paraganglioma (PCPG), and stomach adenocarcinoma (STAD). No changes were found in invasive breast carcinoma (BRCA), cervical squamous cell carcinoma (CESC), lung squamous cell carcinoma (LUSC), pancreatic adenocarcinoma (PAAD), prostate adenocarcinoma (PRAD), sarcoma (SARC), cutaneous skin melanoma (SKCM), or thymoma (THYM). The data are from the UALCAN cancer database [79,80].

**Figure 6 cancers-15-04629-f006:**
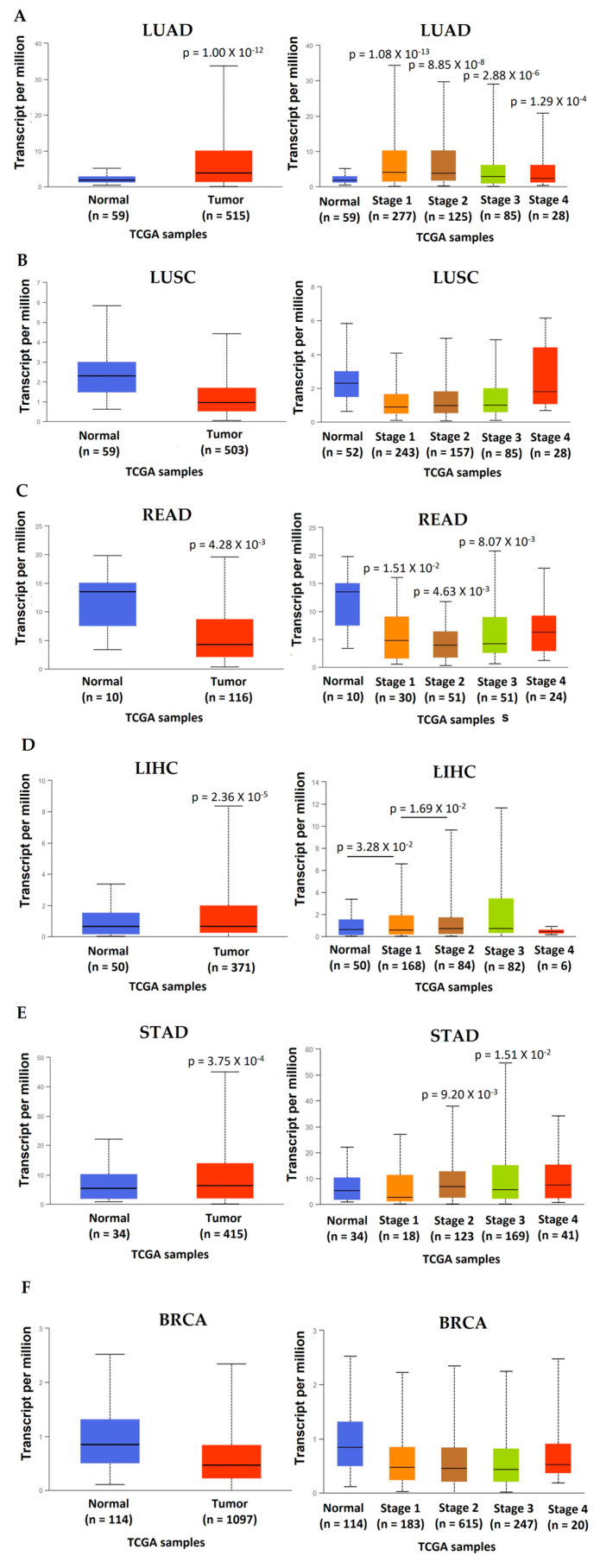
Expression of the *ACHE* gene in lung adenocarcinoma (LUAD), lung squamous cell carcinoma (LUSC), rectum adenocarcinoma (READ), liver hepatocellular carcinoma (LIHC), stomach adenocarcinoma (STAD), and invasive breast carcinoma (BRCA). (**A**) *ACHE* gene expression in LUAD vs. normal samples (*p* = 1.00 × 10^−12^) [79,80] and *ACHE* gene expression in all stages of LUAD vs. normal samples [79,80]. (**B**) *ACHE* gene expression in LUSC vs. normal samples [79,80] and *ACHE* gene expression in all stages of LUSC vs. normal samples [79,80]. (**C**) *ACHE* gene expression in READ vs. normal samples (*p* = 4.28 × 10^−3^) [79,80] and *ACHE* gene expression in all stages of READ vs. normal samples [79,80]. (**D**) *ACHE* gene expression in LIHC vs. normal samples (*p* = 2.36 × 10^−5^) [79,80] and *ACHE* gene expression in all stages of LIHC vs. normal samples [79,80]. (**E**) *ACHE* gene expression in STAD vs. normal samples (*p* = 3.75 × 10^−4^) [79,80] and *ACHE* gene expression in all stages of STAD vs. normal samples [79,80]. (**F**) *ACHE* gene expression in BRCA vs. normal samples [79,80] and *ACHE* gene expression in all stages of BRCA vs. normal samples [79,80].

**Figure 7 cancers-15-04629-f007:**
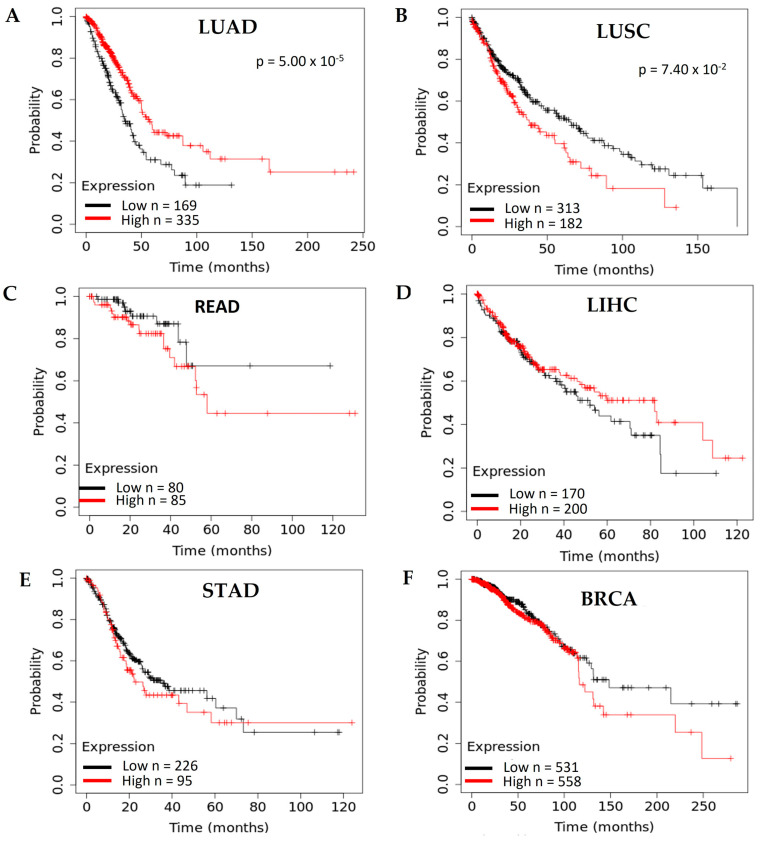
Survival analysis of the *ACHE* gene in lung adenocarcinoma (LUAD), lung squamous cell carcinoma (LUSC), rectum adenocarcinoma (READ), liver hepatocellular carcinoma (LIHC), stomach adenocarcinoma (STAD), and invasive breast carcinoma (BRCA). (**A**) Survival analysis for the *ACHE* gene in LUAD [81]. (**B**) Survival analysis for the *ACHE* gene in LUSC [81]. (**C**) Survival analysis for the *ACHE* gene in READ [81]. (**D**) Survival analysis for the *ACHE* gene in LIHC [81]. (**E**) Survival analysis for the *ACHE* gene in STAD [81]. (**F**) Survival analysis for the *ACHE* gene in BRCA [81].

**Figure 8 cancers-15-04629-f008:**
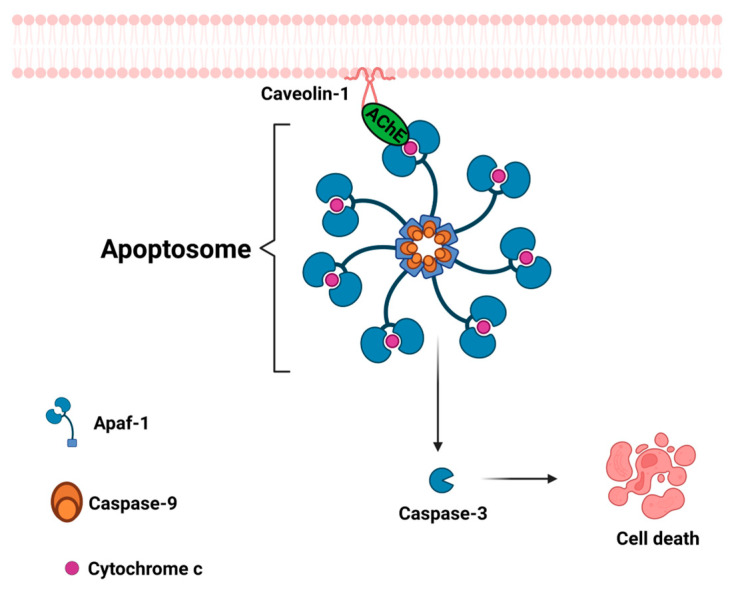
Participation of acetylcholinesterase in apoptosome formation. Acetylcholinesterase binds to caveolin-1, which allows the binding of Apaf-1 to cytochrome C, which consequently causes cleavage of procaspase-9 to produce the active form. Then, caspase 9 promotes the activation of the caspase cascade and apoptosis. Created with Biorender.com (accessed on 12 June 2023).

**Figure 9 cancers-15-04629-f009:**
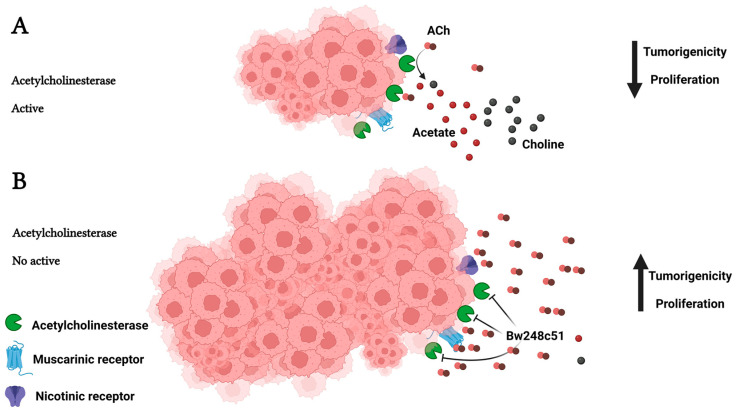
The role of AChE in tumorigenicity. AChE is an essential protein in regulating the ACh levels in non-cholinergic organs, since this neurotransmitter promotes cell proliferation, inhibition of apoptosis, migration, and invasion through acetylcholine receptors (AChRs). (**A**) In normal conditions, AChE controls ACh levels by hydrolyzing this neurotransmitter, preventing the ACh from reaching AChRs; therefore, the tumor growth is controlled, decreasing tumorigenicity and proliferation. (**B**) When AChE is chemically inhibited (the BW248c51 inhibitor is impermeable to the cell membrane) or the AChE activity decreases for another cause, the ACh levels are not modified; therefore, this neurotransmitter reaches AChRs, promoting cell proliferation, inhibiting apoptosis, and increasing migration and invasion. This is reflected in tumor growth, increasing tumorigenicity, proliferation, and metastasis. Created with Biorender.com (accessed on 12 June 2023).

**Figure 10 cancers-15-04629-f010:**
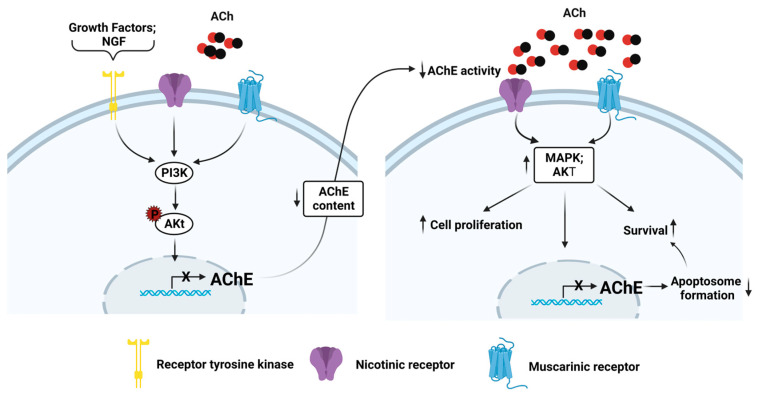
The participation of the PI3K/Akt pathway in the deregulation of AChE. Activation of the RTK receptors, nAChRs, and mAChRs could trigger the PI3K/Akt pathway, decreasing AChE content and AChE activity. The low AChE activity could increase ACh levels that stimulate nAChRs and mAChRs, increasing a sustained hyperactivation of the PI3k/Akt and MAP kinase pathways that promote cell proliferation, survival, and sustained AChE decrement. The AchE decrement could affect the apoptosome formation, increasing survival. Created with Biorender.com (accessed on 12 June 2023).

**Figure 11 cancers-15-04629-f011:**
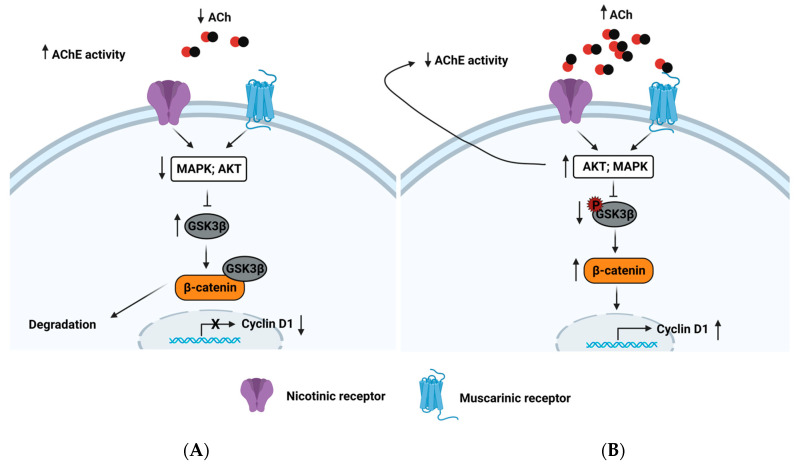
AChE activity regulates cell proliferation in HCC. (**A**) High AChE activity hydrolyzes to ACh, preventing nAChR and mAChR stimulation. These inactive MAPK and Akt pathways cause a reduction in the GSK3β phosphorylation levels, resulting in greater GSK3β activation, β-catenin degradation, and downregulation of cyclin D1. (**B**) A drop in AChE activity promotes high ACh levels, stimulating nAChRs and mAChRs. This induces a triggering of the MAPK and Akt pathways, promoting an increment in the GSK3β phosphorylation levels, resulting in the inactivation of GSK3β, which allows reaching a stable level of β-catenin in the cytosol and translocating to the nucleus. This induces upregulation of cyclin D1. The Akt pathway can also decrease both the AChE content and AChE activity. Created with Biorender.com (accessed on 12 June 2023).

**Figure 12 cancers-15-04629-f012:**
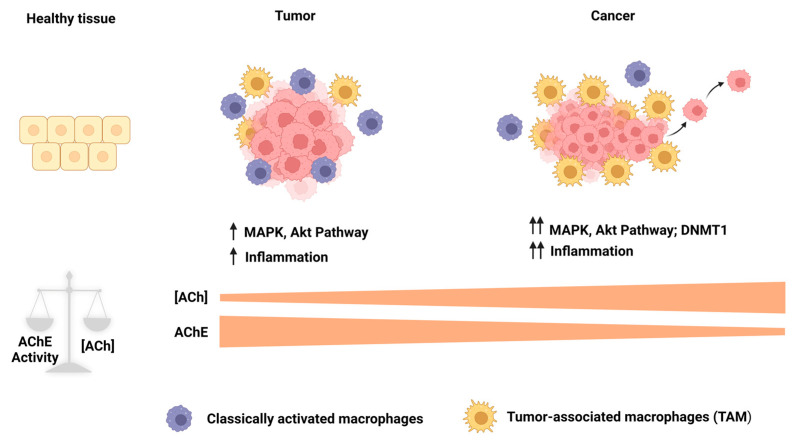
Role of AChE from tumor initiation to progression to metastasis. A balance between ACh levels and AChE activity in normal tissue favors tissue homeostasis. During the initial stages of the tumor, AChE levels increase, causing a decrement in ACh concentrations that affect the tumor microenvironment. This allows both the inflammation and TAMs to increase, favoring tumor development. As the MAPK and Akt pathways are hyperactivated, they induce a decrease in AChE that is also potentiated by DNMT1-mediated hypermethylation in aggressive tumors. A reduction in AChE activity increases ACh levels, directly affecting the tumor and inducing cell proliferation, survival, and metastasis. TAMs: tumor-associated macrophages. Created with Biorender.com (accessed on 12 June 2023).

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
