# Peer review of "Changes in the Acetylcholinesterase Enzymatic Activity in Tumor Development and Progression"

_cancers, 2023, doi:10.3390/cancers15184629_

Round 1
Reviewer 1 Report
Much of the research in the field of AChE (Acetylcholinesterase) and its role in cancer biology is still in its early stages. The relationship between cholinergic signaling, AChE, and cancer is complex and likely varies depending on the specific type of cancer. As such, this area remains an active focus of investigation, warranting further study to gain a comprehensive understanding of its implications for cancer initiation, progression, and metastasis. The authors have made valuable contributions by presenting evidence supporting the role of AChE in tumor initiation, progression, and metastasis.
One primary concern is regarding the novelty of the hypothesis proposed by the authors. It would be beneficial to ascertain whether the authors have reviewed existing literature, particularly a book chapter that may have touched on similar concepts. Understanding the novelty of their manuscript concerning the book chapter will help establish the originality of their work. Moreover, based on its novelty, the manuscript could be considered for publication as either a review paper or an opinion paper.
Additionally, it is essential to improve the quality of the figures to enhance the manuscript's professionalism. Clear and well-designed figures will aid in conveying the information effectively and engaging the readers.
Lastly, the language of the manuscript should be refined to ensure clarity and coherence. A clear and concise writing style will facilitate the communication of complex ideas and findings to the readers more effectively.
The language of the manuscript should be refined to ensure clarity and coherence.
Author Response
Reviewer 1:
Point 1. “One primary concern is regarding the novelty of the hypothesis proposed by the authors. It would be beneficial to ascertain whether the authors have reviewed existing literature, particularly a book chapter that may have touched on similar concepts. Understanding the novelty of their manuscript concerning the book chapter will help establish the originality of their work. Moreover, based on its novelty, the manuscript could be considered for publication as either a review paper or an opinion paper.”
Response 1: We appreciate your comments on our work. Regarding your observation related to the chapter of the book you mentioned, we believe that you are referring to this one: “Acetylcholinesterase and human cancers” (Richbart et al. 2021). We must admit that we were not aware of its existence, for which we appreciate that you told us about it, and of course, we will include it in our appointments. We reviewed the book chapter. In most reviews on AChE and cancer, the authors begin by explaining both the cholinergic system and the description of AChE (the book chapter mentioned does this as well). In our case, unlike the chapter of the book, we gave greater importance to the signaling between ACh and its receptors, since this would better understand our work.
On the other hand, only the first part of our work talks about the low activity of AChE and its relationship with cancer (observations that have been mentioned not only in the chapter of the book but in other previous reviews), so it is impossible set aside these findings in our work. In addition, we describe in more detail a connection between the signaling pathway, the effects of this low enzyme activity and high ACh levels, and how this could play a relevant role in cancer. Even this hypothesis (which is how we stated it in our work) was also raised before us (and the authors of this chapter of the book) by Martínez-Moreno P. et al. (2006), Zhao Y. et al. (2011) and our work group (Pérez-Aguilar B. et al. (2015), which reinforces the idea that one of the explanations for AChE and its relationship with cancer is strongly directed towards that.
The second part of our work focuses on tumors with high enzymatic activity, and we give a possible explanation for their participation in these tumors. As a novelty, we propose that AChE could play a proinflammatory role, which also plays a relevant role in cancer, and we detail this possible participation in some detail. For all these reasons, our review could shed more light on the role of AChE in the development and progression of different types of tumors.
Point 2. “Additionally, it is essential to improve the quality of the figures to enhance the manuscript's professionalism. Clear and well-designed figures will aid in conveying the information effectively and engaging the readers.”
Response 2: The quality of the figures was improved.
Point 3: “Lastly, the language of the manuscript should be refined to ensure clarity and coherence. A clear and concise writing style will facilitate the communication of complex ideas and findings to the
readers more effectively.”
Response 3: The manuscript was subjected to a major English revision and was corrected and improved.
Please see the attachment (Paper)

Reviewer 2 Report
Previous research has shown an association of AChE in breast cancer, where this gene is predominantly deleted. Expression of AChE was found to be reduced in lung cancer. Further analysis from The Cancer Genome Atlas shows that many more cancers results in a reduction of AChE expression. The reviewers also note different degrees of expression in a range of cancers showing variability. It is pretty hard to conclude much from this, other than AChE could be involved in cancer. However, AChE is unlikely to be a main driver in cancer or a drug target.
The argument that AChE is a possible tumor suppressor protein is currently unfounded at present and connection to apoptosis is indirect and not validated experimentally to any significant degree.
It did not help that the review was verbose. It would have been better if this review was much more condensed and focused on the key research papers that implicated AChE in cancer, then shorter sections on correlation studies. Currently, this paper is not suitable as a review paper that would attract a good readership. Enthusiasm for this review is low, and I am doubtful the readership would find this review particularly useful. I have written a series of points below that I also picked up on during my review:
Line 17: try and not to use ‘its’, but say what it is to stop possible confusion: ‘however, controversy has been observed regarding its participation in cancer, since in some tumors’
Line 19: bad grammar ‘and proposes to AChE
Font size on figures should be made larger. Diagrams and graphs had very small font size. Diagrams were also low resolution, so will need to be improved.
The simple summary is longer than the abstract. The simple summary could be written in a more concise manner.
The authors in the abstract indicate that acetylcholinesterase has a non-enzyme effect that is associated with apoptosis. Please explain? Surely the enzyme activity is linked to cell survival? I found the abstract to be cryptic, rather than factual. The authors will need to rewrite the abstract to accurately portray the review.
The association of acetylcholinesterase of apoptosis is indirect consequences, i.e, through the activation of ACh receptors that leads to signal transduction through ERK1/2, Akt, and PKA pathways that enhances cell survival. This could simply be stated, rather than a cryptic abstract.
Line 43: ‘overpowering evidence’, please don’t use the word ‘overpowering’, this is not scientifically appropriate. Perhaps use ‘compelling’ or ‘strong’ instead?
RSK activation of NFkB has not been reported as a main pathway of RSK to drive proliferation. The authors will need to cite paper to back up the link between RSK and NFkB. The connection of RSK to NFkB is not a typically activated pathway from RSK (if at all?). On a similar point, a reference to link ERK to CCND1 is also required.
The review has a detailed description to different spiced forms of AChE at the start of the review, but the relevance to cancer is lacking. If splicing is not involved in cancer development, then this section seems to have little relevance to the review, or could be condensed to a few sentences only. Later in the review, there was reference to the splice forms and cancer. Perhaps the spliced forms should be introduced later on in the review in a concise manner?
The review lacked focused and was too long for what it was trying accomplish. Much editing to reduce the page number to make for a more satisfactory read is required.
Some minor proof reading needed
Author Response
Reviewer 2:
Point 1: “Line 17: try and not to use ‘its’, but say what it is to stop possible confusion: ‘however, controversy has been observed regarding its participation in cancer, since in some tumors’”
Response 1: The manuscript was subjected to a major English revision and was corrected and improved.
Point 2: “Line 19: bad grammar ‘and proposes to AChE.”
Response 2: The manuscript was subjected to a major English revision and was corrected and improved.
Point 3: “Font size on figures should be made larger. Diagrams and graphs had very small font size. Diagrams were also low resolution, so will need to be improved.”
Response 3: The quality of the figures was improved.
Point 4: The simple summary is longer than the abstract. The simple summary could be written in a more concise manner.
The authors in the abstract indicate that acetylcholinesterase has a non-enzyme effect that is associated with apoptosis. Please explain? Surely the enzyme activity is linked to cell survival? I found the abstract to be cryptic, rather than factual. The authors will need to rewrite the abstract to accurately portray the review.
The association of acetylcholinesterase of apoptosis is indirect consequences, i.e, through the activation of ACh receptors that leads to signal transduction through ERK1/2, Akt, and PKA pathways that enhances cell survival. This could simply be stated, rather than a cryptic abstract.”
Response 4: Abstract: Acetylcholinesterase is a well-known protein because of the relevance of its enzymatic activity in the hydrolysis of acetylcholine in nerve transmission. In addition to the catalytic action, it exerts non-catalytic functions; one is associated with apoptosis, where acetylcholinesterase could significantly impact the survival and aggressiveness observed in cancer. The participation of AChE as part of the apoptosome could explain the role in tumors since a drop-in AChE content would increase cell survival due to poor apoptosome assembly. Likewise, the high ACh content caused by the drop-in enzymatic activity could induce cell survival mediated by the overactivation of acetylcholine receptors (AChR) that activate anti-apoptotic pathways. On the other hand, in tumors where high enzymatic activity has been observed, AChE could be playing a different role in the aggressiveness of cancer; in this review, we propose that AChE could have a pro-inflammatory role since the high enzyme content would cause a decrease in ACh, which has also been shown to have anti-inflammatory properties, as discussed in this review. In this review, we analyze the changes that the enzyme could display in different tumors, considering the different levels of regulation that the acetylcholinesterase undergoes, from the control of epigenetic changes in the mRNA expression and changes in the enzymatic activity and its molecular forms. We focused on explaining the relationship between acetylcholinesterase expression and its activity in the biology of various tumors. We present state-of-the-art knowledge regarding this fascinating enzyme that positions as a remarkable target for cancer treatment.
Point 5: “Line 43: ‘overpowering evidence’, please don’t use the word ‘overpowering’, this is not scientifically appropriate. Perhaps use ‘compelling’ or ‘strong’ instead?”
Response 5: We have used strong, thank you.
Point 6: “RSK activation of NF-kB has not been reported as a main pathway of RSK to drive proliferation. The authors will need to cite paper to back up the link between RSK and NF-kB. The connection of RSK to NFkB is not a typically activated pathway from RSK (if at all?). On a similar point, a reference to link ERK to CCND1 is also required.”
Response 6: There are reports supporting the link between RSK and NF-kB (included in the review). It has been shown that p60PSK phosphorylates IkB, which allows the activation of NF-kB. We have highlighted reference 21 in the figure since it describes the activation of NF-kB by ACh and the ACh receptor, activating the ERK/MEKK1/RSK/NF-kB pathway, which increases proliferation, as these authors suggested.
Regarding ERK and CCND1, the citations in the figure supporting the link between Erk and cyclin D1 have already been added. The reference 23 is highlighted since it describes the expression of cyclin D1 through ACh and the ACh receptor.
Point 7: “The review has a detailed description to different spiced forms of AChE at the start of the review, but the relevance to cancer is lacking. If splicing is not involved in cancer development, then this section seems to have little relevance to the review, or could be condensed to a few sentences only. Later in the review, there was reference to the splice forms and cancer. Perhaps the spliced forms should be introduced later on in the review in a concise manner?”
Response 7: Regarding AChE, we believe that it would be impossible to leave out the detailed explanation of alternative splicing since it is needed to understand better the isoforms that can be present in some cancer. In addition, we designed the present manuscript to present general information regarding the enzyme; we considered this section important to gain context for a better understanding of cancer-related issues.
Point 8: “The review lacked focused and was too long for what it was trying accomplish. Much editing to reduce the page number to make for a more satisfactory read is required.”
Response 8: We are convinced that the manuscript is long enough to leave clear the main idea, we have edited some paragraphs, but it is too difficult to cut text and not lose essential ideas.
Please see the attachment (Paper)

Round 2
Reviewer 2 Report
Little had been done to shorten and refocus the review, which is important to do. This review is currently difficult read, as it is too long. Some points below would help to increase the accuracy of the review. I would encourage a higher degree of cutting down and editing, if at all possible, please. There was next to nothing since last review. The authors are trying hard to convince the readership that AChE is a tumor suppressor involved in cancer, yet the evidence is not completely water-tight, I would advise the authors to be unbiased in this review, as much as possible, and talk about the evidence in a way that is factual.
1) Abstract: use a different term: ‘drop-in’ is not an appropriate term. Use words like ‘reduction’, ‘inhibition’, ‘lower’, rather than a term more often used to suggest somebody drops into the office or arrives unexpectedly.
2) Abstract: ‘; in this review, we propose that AChE could have a pro-inflammatory role since the high enzyme content would cause a decrease in ACh, which has also been shown to have anti-inflammatory properties, as discussed in this review’. No need to have 2 uses of ‘review’ words in this sentence.
3) Concerns on accuracy of Figure 1:
MEKK1 is not a downstream target of ERK or an upstream component from RSK, as the diagram suggests.
I think the authors are likely confusing MEKK1 with MEK1. MEKK1 is part of the c-JUN pathway and is upstream of NFkB and is not actually part of the ERK/RSK pathway. The references linked to the pathway indicated in the paper on MEKK1 and NFkB is also not related to the ERK/RSK pathway also. The authors need to consider that their cell signaling diagram is wrong and the text and references are currently miss-leading.
I presume Ca2+ is activating Erk via PKA? Please add PKA to the diagram, as Erk is not directly activated by Ca2+. Also, cAMP is surely also in this pathway involving PKA too?
RSK phosphorylation of IKKB and then NFkB should be in the figure, and also Ref 26 should be added to the figure legend to indicate this signalling input.
4) Line 561: ‘Although it is well known that AChE is a tumor suppressor protein [102,103], it is imperative to elucidate the mechanism that decreases its synthesis and activity.’
I would be careful, AChE is not a ‘well known’ tumor suppressor and the referenced papers only discuss that AChE might be a potential tumor suppressor (only actually described in reference 102, and only mentioned briefly, in 1 sentence). Both 102 and 103 references are more related to the links of AChE with apoptosis. It would be safer to say ‘potential’, as tumor growth type assays to determine whether AChE is a bona fide tumor suppressor have yet to be carried out (as far as I can see from the literature, and based on the references provided in this review paper). The evidence that AChE could be classified as a tumor suppressor is weak at present (further work is needed), However, evidence does suggest that alterations of AChE might augment cancer growth in some circumstances.
Some polishing of sentences is required.
Author Response
Dear reviewer:
General point:
Little had been done to shorten and refocus the review, which is important to do. This review is currently difficult read, as it is too long. Some points below would help to increase the accuracy of the review. I would encourage a higher degree of cutting down and editing, if at all possible, please. There was next to nothing since last review. The authors are trying hard to convince the readership that AChE is a tumor suppressor involved in cancer, yet the evidence is not completely water-tight, I would advise the authors to be unbiased in this review, as much as possible, and talk about the evidence in a way that is factual.
General response:
We have removed some sections and paragraphs of the article to follow the reviewer's recommendations.
Human AChE sequences have three potential N-glycosylation sites [68] and could contain a variable number of oligosaccharides incorporated by post-translational processing, taking in count the complete subunit of AChE has a variable molecular mass between 70 and 80 kDa. In addition, for the same organism, the sugar composition of each AChE-bound oligoglycan frequently varies between different cells (erythrocytes or lymphocytes) and, for the same tissue, between a normal one and a pathological one [69,70]. These differences in the AChE-bound oligoglycans may be helpful for the diagnosis of some diseases [69].
Directed mutagenesis has shown that removing N-glycosylation sites reduces the secretion of the AChE from 20% to 1 % relative to the native type. In any case, the enzymatic activity does not appear to change [71]. In chicken cell cultures, tunicamycin, a potent inhibitor of N-glycosylation, suppresses the formation of the catalytically active enzyme, but this does not affect protein synthesis [59]. Therefore, the amount of inactive AChE not competent for catalysis significantly increases. The correct glycosylation of the AChE subunit is essential for the formation of oligomers and their subsequent secretion [72], as well as to protect them from proteolysis [73]. In some cases, it has been observed that the AChE devoid of carbohydrates or abnormal glycosylation could lose its catalytic activity or address a flawed cell compartment [74]. Therefore, AChE glycosylation is relevant for the protein to achieve optimal folding for catalysis and for each subunit to interact with other catalytic subunits to form homo-oligomers and structural subunits (non-catalytic, PRiMA, and ColQ) to form hetero-oligomers.
Point 1:
Abstract: use a different term: ‘drop-in’ is not an appropriate term. Use words like ‘reduction’, ‘inhibition’, ‘lower’, rather than a term more often used to suggest somebody drops into the office or arrives unexpectedly.
Response 1:
We have changed those words, and we have used reduction and lower:
“The participation of AChE as part of the apoptosome could explain the role in tumors since a lower AChE content would increase cell survival due to poor apoptosome assembly. Likewise, the high ACh content caused by the reduction enzymatic activity could induce cell survival mediated by the overactivation of acetylcholine receptors (AChR) that activate anti-apoptotic pathways.”
Point 2:
Abstract: ‘; in this review, we propose that AChE could have a pro-inflammatory role since the high enzyme content would cause a decrease in ACh, which has also been shown to have anti-inflammatory properties, as discussed in this review’. No need to have 2 uses of ‘review’ words in this sentence.
Response 2:
We have corrected the text and eliminated one “review” word.
“; in this review, we propose that AChE could have a pro-inflammatory role since the high enzyme content would cause a decrease in ACh, which has also been shown to have anti-inflammatory properties.”
Point 3:
Concerns on accuracy of Figure 1:
MEKK1 is not a downstream target of ERK or an upstream component from RSK, as the diagram suggests.
I think the authors are likely confusing MEKK1 with MEK1. MEKK1 is part of the c-JUN pathway and is upstream of NFkB and is not actually part of the ERK/RSK pathway. The references linked to the pathway indicated in the paper on MEKK1 and NFkB is also not related to the ERK/RSK pathway also. The authors need to consider that their cell signaling diagram is wrong and the text and references are currently miss-leading.
I presume Ca2+ is activating Erk via PKA? Please add PKA to the diagram, as Erk is not directly activated by Ca2+. Also, cAMP is surely also in this pathway involving PKA too?
RSK phosphorylation of IKKB and then NFkB should be in the figure, and also Ref 26 should be added to the figure legend to indicate this signalling input.
Response 3:
Thank you for informing us about the error in our diagram in Figure 1A. We have corrected the bug and modified the signaling route. About adding PKA and cAMP to the route, since the articles we are citing do not mention it, based on those articles, we do not believe it is necessary to make the route much larger. Quote 26 has been appended to the figure caption, as the referee proposed.
Point 4:
Line 561: ‘Although it is well known that AChE is a tumor suppressor protein [102,103], it is imperative to elucidate the mechanism that decreases its synthesis and activity.’
I would be careful, AChE is not a ‘well known’ tumor suppressor and the referenced papers only discuss that AChE might be a potential tumor suppressor (only actually described in reference 102, and only mentioned briefly, in 1 sentence). Both 102 and 103 references are more related to the links of AChE with apoptosis. It would be safer to say ‘potential’, as tumor growth type assays to determine whether AChE is a bona fide tumor suppressor have yet to be carried out (as far as I can see from the literature, and based on the references provided in this review paper). The evidence that AChE could be classified as a tumor suppressor is weak at present (further work is needed), However, evidence does suggest that alterations of AChE might augment cancer growth in some circumstances.
Response 4:
We have modified this sentence: Although it is known that AChE protein could be a potential tumor suppressor [95,96], it is imperative to elucidate the mechanism that decreases its synthesis and activity.
Please see the attachment.

Round 3
Reviewer 2 Report
The authors have improved on their text, and have cut down some of the text to help their readership. Importantly, the figures are accurate and clear.
I would have cut down a bit more of the text, but this is a personal choice. I have only one recommendation.
In the abstract please change the words ‘state of art knowledge’ to ‘up-to-date knowledge’, as this would be more appropriate.
Author Response
Dear Reviewer 2:
Point 1:
The authors have improved on their text, and have cut down some of the text to help their readership. Importantly, the figures are accurate and clear.
Response 1:
Thank you for your feedback, it helped to improve the review.
Point 2:
I would have cut down a bit more of the text, but this is a personal choice. I have only one recommendation.
Response 2:
Thank you very much for your insightful comments; we removed as much as we could in the text that did not contribute to the review. We think that the information we leave is adequate for understanding the ideas that we wanted to capture in the paper review.
Point 3:
In the abstract please change the words ‘state of art knowledge’ to ‘up-to-date knowledge’, as this would be more appropriate.
Response 3:
We changed in the abstract the words that the reviewer considered inappropriate, and we have replaced it with the one he suggested.
“…We focused on explaining the relationship between acetylcholinesterase expression and its activity in the biology of various tumors. We present up-to-date knowledge state-of-the-art knowledge regarding this fascinating enzyme that positions as a remarkable target for cancer treatment.”
Please see the attachment.
